# Insights into the genetic diversity and population structure of prevalent *Theileria orientalis* in Bangladesh

Mostak Ahmed[1], Babul Chandra Roy[1], Md Mahfuzur Rahman Sajib[1],
Md Rajiur Rahaman Rabbi[1], Md Makshuder Rahman Zim[1], Md Khalilur Rahman[1],
Md Abu Haris Miah[2], Peru Gopal Biswas[3], Md Hasanuzzaman Talukder[1]*

**1** Department of Parasitology, Bangladesh Agricultural University, Mymensingh, Bangladesh,
**2** Bangladesh Livestock Research Institute, Savar, Dhaka, Bangladesh, **3** Veterinary Office (Food safety and Zoonotic Diseases control), Khulna City Corporation, Khulna, Bangladesh

* talukdermhasan@bau.edu.bd

## Abstract

*Theileria orientalis*, an obligatory intracellular blood protozoon, which causes the substantial economic losses to the cattle industry and is widely prevalent throughout Bangladesh. The study was aimed to assess the nationwide prevalence, genetic diversity and evolutionary divergence of *T. orientalis* parasites in Bangladesh by analyzing bovine blood samples. The genetic characterization, haplotype network analysis and Codon-based evolutionary divergence was conducted by targeting the major piroplasm surface protein (MPSP) gene. The overall prevalence of *T. orientalis* was 63.75% (n = 800), as determined by MPSP-PCR across the eight divisions of Bangladesh with significant regional variation. Subsequent phylogenetic analysis showed that the query MPSP sequences were clustered into two genotypes namely 5 and 7. Genetic diversity indicated high haplotype and moderate nucleotide diversity but consistent with contrasting demographic and selective pressures across regions. The evolutionary divergence analysis revealed the complex genetic landscape of *T. orientalis* population which shaped by historical expansion, ongoing gene flow and localized diversification. The haplotype network analysis identified total 29 distinct haplotypes, highlighting substantial diversity within the two genotypes. These findings demonstrate the occurrence and genetic richness of *T. orientalis* in cattle of Bangladesh. Although the pathogenic impact could not be assessed in this study, the presence of diverse genotypes suggests a need for continued surveillance and future research to determine the clinical and economic relevance of this parasite.

## Introduction

Tick-borne diseases (TBDs) pose a significant threat to the health of domestic and wildlife in the tropical and sub-tropical countries including Bangladesh. Theileriosis is

**Data availability statement:** All data and analysis scripts supporting this study are archived and available at https://doi.org/10.5281/zenodo.17191643.

**Funding:** The authors acknowledge the financial support (Principal investigator, M. H. T.; research and innovation subproject; project code: (RP-C-03-24) from the Livestock and Dairy Development Project (LDDP), jointly funded by the World Bank and Department of Livestock Services (DLS), Ministry of Fisheries and Livestock, Govt. of Bangladesh. The funders had no role in study design, data collection and analysis, decision to publish, or preparation of the manuscript.

**Competing interests:** The authors have declared that no competing interests exist.

a fatal haemoprotozoan tick-borne disease caused by blood protozoa of the genus *Theileria* [1]. Theileriosis leads to significant clinical illness including fever, anemia and death in severe cases and cause considerable economic losses through reduced milk yield, poor weight gain, high treatment costs and cattle mortality [2,3]. Several species of *Theileria* are associated with bovine theileriosis including *T. orientalis*, *T. annulata*, *T. parva*, *T. taurotragi* and *T. velifera* [4,5]. Among these, *T. annulata* and *T. parva* are highly pathogenic, inducing lymphoproliferative diseases known as tropical theileriosis and East Coast fever respectively [6]. In contrast, *T. orientalis* does not induce lymphoproliferation. *T. orientalis* is principally transmitted via ixodid tick belongs to the genus *Haemaphysalis* [7, 8]. Lice, mosquitoes and biting flies may also transmit *T. oreintalis* reported in Australia [9]. Recovered animals often remain lifelong asymptomatic carriers, maintaining low-level parasitaemia for at least 30 months [10]. Although *T. orientalis* usually causes less mortality than *T. annulata* and *T. parva*, it can still be fatal and has been linked to major livestock losses in the past [11–14]. The annual economic impact is estimated at USD 100 million in Japan and Korea, and AUD 20 million in Australia, reflecting high per-animal losses for both dairy and beef producers [15–17]. High molecular prevalence of *T. orientalis* has been reported in several regions including Mymensingh (55.2%), Sirajganj (7.7%), Rangpur (20.4%), Bandarban (68.57%), Khagrachari (42.86%), Bogura (50%), and Jhenaidah (24.13%) of Bangladesh [18–20]. In contrast, several clinical outbreaks of oriental theileriosis have also been reported 70% and 93.3% in India and other countries, raising concerns about the potential for similar events in Bangladesh [21,22]. However, no clinical oriental theileriosis has yet been addressed in Bangladesh. Molecular surveys also indicate that *T. orientalis* is widespread globally, with the prevalence 15% in Pakistan, 36.2% in Myanmar, 36.5% in Thailand, 49.76% in Malaysia, 36.5% in China, 64.8% in Japan, 41.3% in Korea, 8.8% in Egypt and 13% in in cattle of Australia, respectively [23–31]. The global climate changes may has led to the expansion of tick populations in a region where they were not previously found [32]. Treatment failure with buparvaquone, alongside molecular evidence of resistance, suggest that drug efficacy is being compromised in the endemic regions [33,34]. Moreover, tick control strategies has also been challenged due to increasing acaricides resistance [35]. Therefore, theileriosis is becoming increasingly difficult to manage, highlighting the urgent need to explore molecular surveillance of circulating genotypes, monitor genetic diversity and investigate novel therapeutic and integrated control strategies.

Among various diagnostic techniques, the giemsa-stained blood smears and lymph node needle biopsy smears are being the most commonly used standard methods for diagnosis of theileriosis in acute infections. However, these approaches are ineffective and less sensitive for morphological identification of *Theileria* species in carrier healthy animals due to the low parasitaemia. On the other hands, serological methods are effective to detect the antibodies in subclinical theileriosis but shows non specificity due to cross-reactivity or weakening of specific immune responses [36,37]. Therefore, a sensitive and highly specific approches is required for accurate diagnosis of oriental theileriosis. The molecular tool (polymerase chain reaction, PCR) demonstrates the

greater sensitivity, accuracy for detecting *T. orientalis* in carrier cattle with low parasitemia [38–40]. In addition, molecular techniques also allow to detect parasite DNA, confirming the presence of parasites in animal at the time of sampling [41,42]. Various molecular markers have been employed to characterize the *T. orientalis* complex, including the small subunit (SSU) of nuclear ribosomal RNA (18S rRNA), the internal transcribed spacers-1&2 (ITS-1 and ITS-2) of nuclear ribosomal DNA, the cytochrome c oxidase III gene, the 23-kDa piroplasm membrane protein (p23) gene and the major piroplasm surface protein (MPSP) gene [17,24,43–45]. Among these, the MPSP gene serves as a useful marker in epidemiological and phylogenetic studies of *T. orientalis* across various geographic regions [23,35,46]. Bovine hemoprotozoan parasites often develop antigenic polymorphisms, enabling them to evade host immune defences and establish persistent infections [47]. Similar findings in *T. orientalis* highlight the role of the major piroplasm surface protein (MPSP) as the principal marker of antigenic variation [48]. It encodes an immunodominant surface protein expressed on the piroplasm stage of infected erythrocytes, which plays an important role in host–parasite interactions [23,26]. Importantly, variations in MPSP genotypes have also been associated with different levels of pathogenicity, with some genotypes implicated in clinical disease outbreaks, thereby highlighting its significance in understanding disease severity and immune evasion [49]. Hence, the MPSP gene provides an important molecular target for diagnostic tools, surveillance programs, and potential control measures for oriental theileriosis. The majority of research on *T. orientalis* in Bangladesh focuses on localized studies, such as individual outbreaks, regional prevalence or clinical investigations with limited comprehensive national data on prevalence. The genetic diversity of *T. orientalis* has exclusively been lacked to address the population structure and evolutionary dynamics of the circulating strains in the country. These creates a considerable gap in understanding the geographic spread of *T. orientalis* genotypes across different bio-ecological zones of the country. Additionally, a nationwide study is essential to identify the most prevalent and virulent strains which would facilitate the development of better diagnostic tools, vaccines and treatment approaches. Therefore, the present study was aimed to investigate the prevalence of emerging oriental theileriosis along with the genetic diversity and population structures among the cattle population of Bangladesh.

## Methods

### Ethical declaration

The study protocol was reviewed and approved by the Animal Welfare and Experimentation Ethics Committee (AWEEC) for experimentation on Animal, Birds, Microbes and Living Natural Sources, Bangladesh Agricultural University (BAU), Mymensingh, Bangladesh. In addition, informed consent was obtained from cattle owners prior to field sample collection. Ethical approval for the study was also obtained from the Animal Welfare and Experimentation Ethics Committee of Bangladesh Agricultural University (BAU), Mymensingh (AWEEC/BAU/2023(2)/14(a)).

### Study area and sample collection

For the nationwide surveillance, the research activities were conducted from November 2022 to October 2024. Samples were collected randomly from cattle keeping by the subsistent farmers in the villages as well as the farms belongs to the producer groups (PG) under the LDDP project, Department of Livestock Services, across the eight divisions of Bangladesh: namely, Rangpur, Rajshahi, Dhaka, Mymensingh, Chattogram, Khulna, Sylhet and Barishal (Fig 1). A total of 800 blood samples were collected, with an equal number of cattle (n = 100) sampled from each of the eight administrative divisions. Sampling was conducted irrespective of age, breed, or sex to capture a broad and diverse representation of the cattle population. Animals were listed in a numeric sequence from each herd and chosen based on a series of random numbers produced by a computer. Approximately 2 ml blood were collected aseptically from the jugular vein and transferred into EDTA-containing NMC vacuum tubes and mixed thoroughly with EDTA by eight note motion and kept in a chilled box. Each blood sample was properly labelled and shipped to the laboratory in the Department of Parasitology, Bangladesh Agricultural University and stored at −20°C for further use.

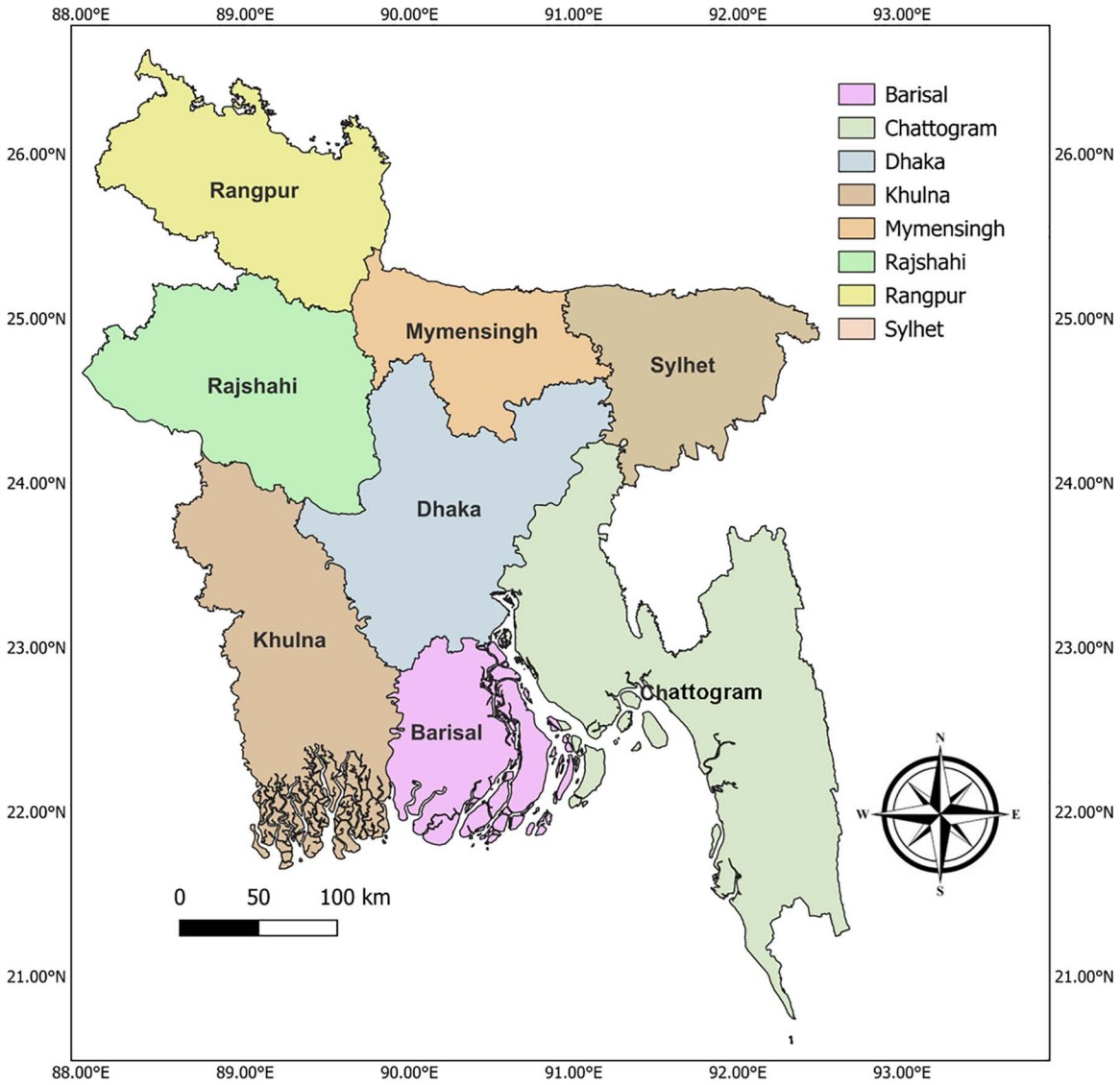

**Fig 1. Geographical distribution of the eight divisions of Bangladesh where tick-borne bovine oriental theileriosis was investigated.**

## Preparation of Genomic DNA

Genomic DNA was extracted from collected blood samples using commercially available DNA extraction Kit (Bio Basic, Canada) following the instructions of the manufacturer. The concentration of the extracted DNA was measured using a Nano drop spectrophotometer at the 260/280 and 260/230 ratios (Bio-Rad Laboratories Inc., Hercules, CA, USA)) and stored at −20°C for subsequent analysis.

## MPSP gene amplification and sequencing

All DNA samples were screened for *T. orientalis* through PCR amplification of the Major piroplasm surface protein (MPSP) gene. The PCR was performed in a final reaction volume of 25 μL comprising 12.5 μL of 2 × master mix (TaKaRa, Japan), 0.5 μL of each primer (20 pmol/μL), 8.5 μL nuclease-free double-distilled water and 3.0 μL of genomic DNA template. The

primers TOMPSP-F (5′-CTTTGCCTAGGATACTTCCT-3′) and TOMPSP-R (5′-ACGGCAAGTGGTGAGAACT-3′) were custom synthesized and employed in PCR to amplify a 776-bp section of the *T. orientalis* MPSP gene [23]. The thermal conditions included an initial denaturation at 94°C for 10 minutes, followed by 35 cycles of denaturation at 94°C for 1 minute, annealing at 58°C for 1 minute and extension at 72°C for 1 minute. A final extension of the strands was given at 72°C for 10 minutes. PCR products were separated on 1.5% agarose gels, stained with Midori Green Advance DNA Stain (NIPPON Genetics, Europe) and visualized under UV illumination using an Alpha Imager (Bio-Rad, USA). All PCR reactions were performed with a negative control to verify the absence of contamination. In this study, the MPSP primers were designed to target a genomic region unique to the *T. orientalis* complex. Despite the presence of homologous sequences in *T. annulata* and *T. parva*, prior validation studies have confirmed that this primer pair exclusively amplifies *T. orientalis*. As DNA from other *Theileria* species is not present in our local context, the assay is considered highly specific. A total of 32 (four from each division) high quality MPSP-positive PCR products were purified and sequenced commercially by GeneCreate Biotech, China [50]. The newly generated nucleotide sequences from this study were deposited in the Gen-Bank database and can be retrieved under the accession numbers PV568095 to PV568107 and PX310829 to PX310847, respectively.

**Similarity indices, phylogenetic positioning and genetic diversity analysis of MPSP gene**

A total of 32 sequences were obtained in this study and subsequently subjected to BLAST analysis (https://blast.ncbi.nlm.nih.gov/Blast.cgi) in the NCBI database, and their identities and similarities were compared with *T. orientalis* reference sequences available in GenBank. Initially, a similarity matrix was generated using the Sequence Identity and Similarity (SIAS) web tool (imed.med.ucm.es/Tools/sias.html), which calculates pairwise similarity scores between the sequences. Subsequently multiple sequence alignment was performed using ClustalW in MEGA version 11 [51] followed by trimming of the obtained sequences from the GenBank. Phylogenetic analysis was performed through Maximum Likelihood (ML) method to detect the closely related genotypes of *T. orientalis* by keeping the *Babesia bigemina* sequence (M85187.1) as out-group. The phylogenetic tree was validated by running the analysis on 1000 bootstrapped input datasets and cross-referencing it against the Tamura-Nei substitution model [52]. The 32 query sequences (four from each division) were used for subsequent genetic analysis. Population nucleotide diversity indices such as nucleotide diversity per site ($\pi$), average number of nucleotide differences (k), mean genetic diversity (Hd), genetic differentiation parameters (Fst and Nm) and neutrality tests including Tajima's D and Fu Li's F test were calculated using DnaSP ver. 6.12.03 [53]. Codon-based evolutionary divergence estimation for the nucleotide sequences (32) of the *T. orientalis* MPSP gene were calculated with MEGA 11 [51]. Initially, sequences were aligned using MUSCLE and thereafter the Nei-Gojobori (Jukes-Cantor) method was chosen, and pairwise deletion was applied to handle gaps or missing data.

**Haplotype network analysis**

MUSCLE algorithm within MEGA 11 was employed to align the gene sequences of the MPSP gene. Aligned sequences were subsequently subjected to haplotype network analysis using DNA Sequence Polymorphism software (DnaSP v.6.12), as outlined by Rozas et al. 2017 [54]. Visualization of haplotypes was achieved using a median joining network (MJN) facilitated by the PopART program [55,56]. The MJN analysis was constructed using 32 generated sequences, comprising four sequences from each division.

**Statistical analysis**

Pearson's chi-square and Fisher's exact test were used to assess the association of TBP detection rates in different study locations using IBM SPSS Statistics (Version 27) [57]. A p-value was considered significant when it was < 0.05.

## Results

### Molecular prevalence of *T. orientalis* infection of cattle in Bangladesh

The prevalence of *T. orientalis* infection in cattle was determined by PCR followed by gel electrophoresis (S1 Fig). A total of 510 bovine blood was found positive with *T. orientalis* indicating the overall prevalence 63.75% (510/800). All the positive blood samples collected from 8 divisions of the country showed a significant variation (p < 0.001) of *T. orientalis* infection. High *T. orientalis* infection rate was recorded in cattle of Chattogram division (84.00%), followed by Barishal (82.00%), Sylhet (76.00%), Mymensingh (74%), Khulna (64.00%), Rajshahi (54.00%), Rangpur (40.00%) and Dhaka (36.00%) divisions, respectively (Table 1).

### Analyses of MPSP gene sequence similarity

The identity values of newly generated MPSP gene sequences were varied from 88.41 to 98.02% with the reference sequences (Table 2). The similarity of identity among the query sequences ranges from 88.93 to 100% (Table 3).

### Phylogenetic positioning

Phylogenetic analysis of the *T. orientalis* MPSP gene sequences revealed that 24 sequences were clustered within genotype 5, forming a strongly supported clade together with the Indian isolates (PV55977.1). Within this lineage, the sequences generated in this study were subdivided into several distinct sub-branches, forming monophyletic groups that reflect considerable intra-genotypic diversity. In addition, eight sequences were assigned to genotype 7, which formed a well-supported sister clade with the isolates of India (HQ444179.1), Bangladesh (OQ144964.1), and Sri Lanka (AB701457.1). The Bangladeshi sequences within this cluster also segregated into well-defined sub-branches, further supporting the presence of intra-genotypic divergence (Fig 2).

### Genetic diversity of *T. orientalis*

A total of 32 sequences from eight divisions (four per division) were analysed, revealing 29 haplotypes and very high overall haplotype diversity (Hd = 0.994 ± 0.009) with moderate nucleotide diversity (π = 0.095 ± 0.007), indicating substantial genetic variation among *T. orientalis* isolates in Bangladesh. Most divisions, including Mymensingh, Chattogram, Barishal, Rangpur and Dhaka showed maximum haplotype diversity (Hd = 1), while Rajshahi, Sylhet, and Khulna had slightly lower diversity (Hd = 0.833), reflecting some shared haplotypes. The number of segregating sites (S) ranged from 2 in Khulna to 177 in Chattogram and the average number of nucleotide differences (K) varied from 1.167 in Khulna to 97.83 in Chattogram, highlighting differences in genetic divergence across the regions. The nucleotide diversity (π) was highest in

Table 1. Prevalence of *T. orientalis* in cattle across all divisions of Bangladesh.

| Location | No. of positive cases | % detection | 95% CI (%) | p-value |
|---|---|---|---|---|
| Dhaka | 36/100 | 36 | 26.6–45.4 | <0.001 |
| Rangpur | 40/100 | 40 | 30.4–49.6 | 0.046 |
| Rajshahi | 54/100 | 54 | 44.2–63.8 | 0.423 |
| Khulna | 64/100 | 64 | 54.6–73.4 | 0.005 |
| Mymensingh | 74/100 | 74 | 65.4–82.6 | <0.001 |
| Sylhet | 76/100 | 76 | 67.6–84.4 | <0.001 |
| Barishal | 82/100 | 82 | 74.5–89.5 | <0.001 |
| Chattogram | 84/100 | 84 | 76.8–91.2 | <0.001 |
| Total | 510/800 | 63.75 | 60.4–67.1 | N/A |

N/A: Not applicable.

**Table 2. BLAST (Basic Local Alignment Search Tool) analysis of *Theileria orientalis* MPSP gene sequences from cattle in Bangladesh showing query accession numbers, genotypes, closest GenBank matches, identity percentages, and countries of origin.**

| Division | Query Sequence (Accession no.) | Genotype | Most identical Sequence (Accession no.) | Identity % | Country |
|---|---|---|---|---|---|
| Rajshahi | PV568095 | Genotype 5 | KU886281.1 | 98.02 | Thailand |
| Rajshahi | PV568096 | Genotype 5 | AB016278.1 | 95.00 | Japan |
| Rajshahi | PX310830 | Genotype 7 | AB701457.1 | 96.44 | Sri Lanka |
| Rajshahi | PX310831 | Genotype 7 | AB701457.1 | 96.44 | Sri Lanka |
| Chattogram | PV568097 | Genotype 5 | OP839193.1 | 97.87 | Malawi |
| Chattogram | PV568098 | Genotype 5 | AB560817.1 | 88.41 | Vietnam |
| Chattogram | PX310834 | Genotype 7 | AB701457.1 | 96.71 | Sri Lanka |
| Chattogram | PX310841 | Genotype 5 | AB701450.1 | 97.93 | Sri Lanka |
| Khulna | PV568099 | Genotype 5 | KU886286.1 | 91.45 | Thailand |
| Khulna | PX310842 | Genotype 5 | PV755977.1 | 91.84 | India |
| Khulna | PX310843 | Genotype 5 | PV755977.1 | 92.11 | India |
| Khulna | PX310844 | Genotype 5 | PV755977.1 | 92.84 | India |
| Mymensingh | PV568100 | Genotype 5 | KU886281.1 | 97.10 | Thailand |
| Mymensingh | PX310836 | Genotype 5 | KU886281.1 | 96.71 | Thailand |
| Mymensingh | PX310837 | Genotype 5 | KU886281.1 | 96.31 | Thailand |
| Mymensingh | PX310829 | Genotype 7 | AB701457.1 | 96.84 | Sri Lanka |
| Dhaka | PV568101 | Genotype 5 | AB871333.1 | 91.83 | Myanmer |
| Dhaka | PV568103 | Genotype 5 | KU886281.1 | 96.18 | Thailand |
| Dhaka | PV568104 | Genotype 5 | PV090972.1 | 89.35 | India |
| Dhaka | PV568105 | Genotype 7 | AB701457.1 | 96.05 | Sri Lankan |
| Sylhet | PV568102 | Genotype 5 | AB871333.1 | 96.97 | Myanmar |
| Sylhet | PX310847 | Genotype 5 | AB571967.1 | 96.71 | China |
| Sylhet | PX310832 | Genotype 7 | AB701457.1 | 96.57 | Sri Lanka |
| Sylhet | PX310833 | Genotype 7 | AB701457.1 | 96.57 | Sri Lanka |
| Rangpur | PV568106 | Genotype 5 | AB871353.1 | 96.97 | Myanmar |
| Rangpur | PX310838 | Genotype 5 | AB871353.1 | 96.97 | Myanmar |
| Rangpur | PX310839 | Genotype 5 | AB871353.1 | 96.10 | Myanmar |
| Rangpur | PX310840 | Genotype 5 | AB871353.1 | 96.97 | Myanmar |
| Barishal | PV568107 | Genotype 5 | PV090972.1 | 89.61 | India |
| Barishal | PX310845 | Genotype 5 | PV775927.1 | 96.97 | India |
| Barishal | PX310846 | Genotype 5 | PV775927.1 | 96.97 | India |
| Barishal | PX310835 | Genotype 7 | AB701457.1 | 96.84 | Sri Lanka |

Chattogram (0.120±0.04) and Rajshahi (0.112±0.029) and lowest in Rangpur (0.003±0.001) and Khulna (0.001±0.004). Neutrality tests were mostly negative or non-significant, consistent with population expansion or purifying selection except in Sylhet, where significantly positive Tajima's D (2.268, $p < 0.05$) and Fu-Li's F (2.432, $p < 0.01$) suggested balancing selection or population contraction. In Chattogram, the highly significant negative Tajima's D ($-0.89133$, $p < 0.001$) indicates an excess of low-frequency polymorphisms, consistent with purifying selection or recent population expansion. The negative but non-significant Fu and Li's F ($-0.85158$) suggests that singleton mutations are not sufficiently frequent to reach statistical significance, reflecting the differing sensitivities of these neutrality tests. Overall, these results indicate considerable haplotype diversity, variable nucleotide divergence and regional differences in sequence evolution, with the *T. orientalis* population largely evolving under neutral or purifying selection (Table 4).

The pairwise comparisons among *T. orientalis* populations from eight divisions of Bangladesh revealed moderate to high genetic differentiation between certain populations. Most Fst values are low to moderate (0–0.587), indicating partial genetic structuring, while some pairs show high differentiation, particularly Rangpur–Khulna (Fst = 0.9619) and Barishal–Rangpur/Khulna (Fst ≈ 0.586–0.588), suggesting these populations are genetically distinct. Negative or near-zero Nst values in several comparisons (Mymensingh–Sylhet, Mymensingh–Chattogram) indicate low sequence divergence and shared haplotypes, possibly due to gene flow or recent common ancestry. The average number of nucleotide differences (Kxy) and nucleotide divergence (Dxy) vary across populations, with the highest Kxy observed between Chattogram–Barishal (99) and the lowest between Khulna–Rangpur (52.5), reflecting differences in sequence divergence. The net nucleotide differences (Da) are further highlight distinct populations, with Rangpur showing the highest divergence from Khulna (Da = 0.06653), whereas some pairs, such as Mymensingh–Sylhet (Da = −0.0038), indicate negligible differentiation (Table 5).

## Codon-based evolutionary divergence

The pairwise codon-based comparisons of the 32 *T. orientalis* sequences reveal two distinct genotypes with contrasting evolutionary patterns. Genotype 5 sequences show very low divergence within and across divisions (0.000–0.372),

**Table 3. Intra-sequences (query) similarity matrix.**

| PV568100 | 100% | | | | | | | | | | | | | | |
|---|---|---|---|---|---|---|---|---|---|---|---|---|---|---|---|
| PX310836 | 100% | 100% | | | | | | | | | | | | | |
| PX310837 | 100% | 100% | 100% | | | | | | | | | | | | |
| PV568095 | 92.81% | 92.81% | 92.81% | 100% | | | | | | | | | | | |
| PV568096 | 92.81% | 92.81% | 92.81% | 100% | 100% | | | | | | | | | | |
| PV568102 | 94.86% | 94.86% | 94.86% | 96.87% | 96.87% | 100% | | | | | | | | | |
| PX310847 | 94.86% | 94.86% | 94.86% | 96.87% | 96.87% | 100% | 100% | | | | | | | | |
| PV568097 | 96.24% | 96.24% | 96.24% | 91.50% | 91.50% | 93.34% | 93.34% | 100% | | | | | | | |
| PV568098 | 96.24% | 96.24% | 96.24% | 91.50% | 91.50% | 93.34% | 93.34% | 100% | 100% | | | | | | |
| PX310841 | 96.24% | 96.24% | 96.24% | 91.50% | 91.50% | 93.34% | 93.34% | 100% | 100% | 100% | | | | | |
| PV568107 | 92.32% | 92.32% | 92.32% | 91.46% | 91.46% | 91.27% | 91.27% | 90.84% | 90.84% | 90.84% | 100% | | | | |
| PX310845 | 92.32% | 92.32% | 92.32% | 91.46% | 91.46% | 91.27% | 91.27% | 90.84% | 90.84% | 90.84% | 100% | 100% | | | |
| PX310846 | 92.32% | 92.32% | 92.32% | 91.46% | 91.46% | 91.27% | 91.27% | 90.84% | 90.84% | 90.84% | 100% | 100% | 100% | | |
| PV568106 | 94.76% | 94.76% | 94.76% | 90.05% | 90.05% | 92.25% | 92.25% | 92.78% | 92.78% | 92.78% | 89.59% | 89.59% | 89.59% | 100% | |
| PX310838 | 94.76% | 94.76% | 94.76% | 90.05% | 90.05% | 92.25% | 92.25% | 92.78% | 92.78% | 92.78% | 89.59% | 89.59% | 89.59% | 100% | 100% |
| PX310839 | 94.76% | 94.76% | 94.76% | 90.05% | 90.05% | 92.25% | 92.25% | 92.78% | 92.78% | 92.78% | 89.59% | 89.59% | 89.59% | 100% | 100% |
| PX310840 | 94.76% | 94.76% | 94.76% | 90.05% | 90.05% | 92.25% | 92.25% | 92.78% | 92.78% | 92.78% | 89.59% | 89.59% | 89.59% | 100% | 100% |
| PV568099 | 90.57% | 90.57% | 90.57% | 88.93% | 88.93% | 89.16% | 89.16% | 89.03% | 89.03% | 89.03% | 89.82% | 89.82% | 89.82% | 93.01% | 93.01% |
| PX310842 | 90.57% | 90.57% | 90.57% | 88.93% | 88.93% | 89.16% | 89.16% | 89.03% | 89.03% | 89.03% | 89.82% | 89.82% | 89.82% | 93.01% | 93.01% |
| PX310843 | 90.57% | 90.57% | 90.57% | 88.93% | 88.93% | 89.16% | 89.16% | 89.03% | 89.03% | 89.03% | 89.82% | 89.82% | 89.82% | 93.01% | 93.01% |
| PX310844 | 90.57% | 90.57% | 90.57% | 88.93% | 88.93% | 89.16% | 89.16% | 89.03% | 89.03% | 89.03% | 89.82% | 89.82% | 89.82% | 93.01% | 93.01% |
| PV568101 | 95.12% | 95.12% | 95.12% | 92.12% | 92.12% | 94.13% | 94.13% | 93.08% | 93.08% | 93.08% | 94.13% | 94.13% | 94.13% | 92.25% | 92.25% |
| PV568103 | 95.12% | 95.12% | 95.12% | 92.12% | 92.12% | 94.13% | 94.13% | 93.08% | 93.08% | 93.08% | 94.13% | 94.13% | 94.13% | 92.25% | 92.25% |
| PV568104 | 95.12% | 95.12% | 95.12% | 92.12% | 92.12% | 94.13% | 94.13% | 93.08% | 93.08% | 93.08% | 94.13% | 94.13% | 94.13% | 92.25% | 92.25% |
| PX310829 | 100% | 100% | 100% | 92.81% | 92.81% | 94.86% | 94.86% | 96.24% | 96.24% | 96.24% | 92.32% | 92.32% | 92.32% | 94.76% | 94.76% |
| PX310830 | 92.81% | 92.81% | 92.81% | 100% | 100% | 96.87% | 96.87% | 91.50% | 91.50% | 91.50% | 91.46% | 91.46% | 91.46% | 90.05% | 90.05% |
| PX310831 | 92.81% | 92.81% | 92.81% | 100% | 100% | 96.87% | 96.87% | 91.50% | 91.50% | 91.50% | 91.46% | 91.46% | 91.46% | 90.05% | 90.05% |
| PX310832 | 94.86% | 94.86% | 94.86% | 96.87% | 96.87% | 100% | 100% | 93.34% | 93.34% | 93.34% | 91.27% | 91.27% | 91.27% | 92.25% | 92.25% |
| PX310833 | 94.86% | 94.86% | 94.86% | 96.87% | 96.87% | 100% | 100% | 93.34% | 93.34% | 93.34% | 91.27% | 91.27% | 91.27% | 92.25% | 92.25% |
| PX310834 | 96.24% | 96.24% | 96.24% | 91.50% | 91.50% | 93.34% | 93.34% | 100% | 100% | 100% | 90.84% | 90.84% | 90.84% | 92.78% | 92.78% |
| PX310835 | 92.32% | 92.32% | 92.32% | 91.46% | 91.46% | 91.27% | 91.27% | 90.84% | 90.84% | 90.84% | 100% | 100% | 100% | 89.59% | 89.59% |
| PV568105 | 95.12% | 95.12% | 95.12% | 92.12% | 92.12% | 94.13% | 94.13% | 93.08% | 93.08% | 93.08% | 94.13% | 94.13% | 94.13% | 92.25% | 92.25% |
| | PV568100 | PX310836 | PX310837 | PV568095 | PV568096 | PV568102 | PX310847 | PV568097 | PV568098 | PX310841 | PV568107 | PX310845 | PX310846 | PV568106 | PX310838 |

indicating high sequence conservation and limited intra-genotypic variability, although some sequences from geographically distant populations exhibit slightly higher divergence, reflecting regional variation. In contrast, Genotype 7 sequences exhibit substantially higher pairwise divergence (0.434–0.659), suggesting the greater genetic heterogeneity and more pronounced population structuring compared to the genotype 5 (Table 6).

## Haplotype network analysis

The median-joining networking of 29 haplotypes revealed a high level of diversity. Hap_26 is emerged as the major ancestral or central haplotype, connecting multiple haplotypes from different regions, while Hap_9 and Hap_28 also served as secondary hubs, indicating sub-structuring within the population (Fig 3). Dhaka and Rangpur displayed the greatest diversity, with haplotypes widely distributed across the network, suggesting their role as genetic mixing centers, whereas haplotypes from Mymensingh, Sylhet and Khulna harboured more unique or peripheral haplotypes, pointing to localized lineage evolution. The clustering of Chattogram and Barishal haplotypes around distinct nodes further reflected regional structuring, although shared haplotypes across multiple regions highlighted evidence of gene flow. The overall star-like expansion

| PX310839 | PX310840 | PV568099 | PX310842 | PX310843 | PX310844 | PV568101 | PV568103 | PV568104 | PX310829 | PX310830 | PX310831 | PX310832 | PX310833 | PX310834 | PX310835 | PV568105 |
|---|---|---|---|---|---|---|---|---|---|---|---|---|---|---|---|---|
| 100% | | | | | | | | | | | | | | | | |
| 100% | 100% | | | | | | | | | | | | | | | |
| 93.01% | 93.01% | 100% | | | | | | | | | | | | | | |
| 93.01% | 93.01% | 100% | 100% | | | | | | | | | | | | | |
| 93.01% | 93.01% | 100% | 100% | 100% | | | | | | | | | | | | |
| 93.01% | 93.01% | 100% | 100% | 100% | 100% | | | | | | | | | | | |
| 92.25% | 92.25% | 90.61% | 90.61% | 90.61% | 90.61% | 100% | | | | | | | | | | |
| 92.25% | 92.25% | 90.61% | 90.61% | 90.61% | 90.61% | 100% | 100% | | | | | | | | | |
| 92.25% | 92.25% | 90.61% | 90.61% | 90.61% | 90.61% | 100% | 100% | 100% | | | | | | | | |
| 94.76% | 94.76% | 90.57% | 90.57% | 90.57% | 90.57% | 95.12% | 95.12% | 95.12% | 100% | | | | | | | |
| 90.05% | 90.05% | 88.93% | 88.93% | 88.93% | 88.93% | 92.12% | 92.12% | 92.12% | 92.81% | 100% | | | | | | |
| 90.05% | 90.05% | 88.93% | 88.93% | 88.93% | 88.93% | 92.12% | 92.12% | 92.12% | 92.81% | 100% | 100% | | | | | |
| 92.25% | 92.25% | 89.16% | 89.16% | 89.16% | 89.16% | 94.13% | 94.13% | 94.13% | 94.86% | 96.87% | 96.87% | 100% | | | | |
| 92.25% | 92.25% | 89.16% | 89.16% | 89.16% | 89.16% | 94.13% | 94.13% | 94.13% | 94.86% | 96.87% | 96.87% | 100% | 100% | | | |
| 92.78% | 92.78% | 89.03% | 89.03% | 89.03% | 89.03% | 93.08% | 93.08% | 93.08% | 96.24% | 91.50% | 91.50% | 93.34% | 93.34% | 100% | | |
| 89.59% | 89.59% | 89.82% | 89.82% | 89.82% | 89.82% | 94.13% | 94.13% | 94.13% | 92.32% | 91.46% | 91.46% | 91.27% | 91.27% | 90.84% | 100% | |
| 92.25% | 92.25% | 90.61% | 90.61% | 90.61% | 90.61% | 100% | 100% | 100% | 95.12% | 92.12% | 92.12% | 94.13% | 94.13% | 93.08% | 94.13% | 100% |

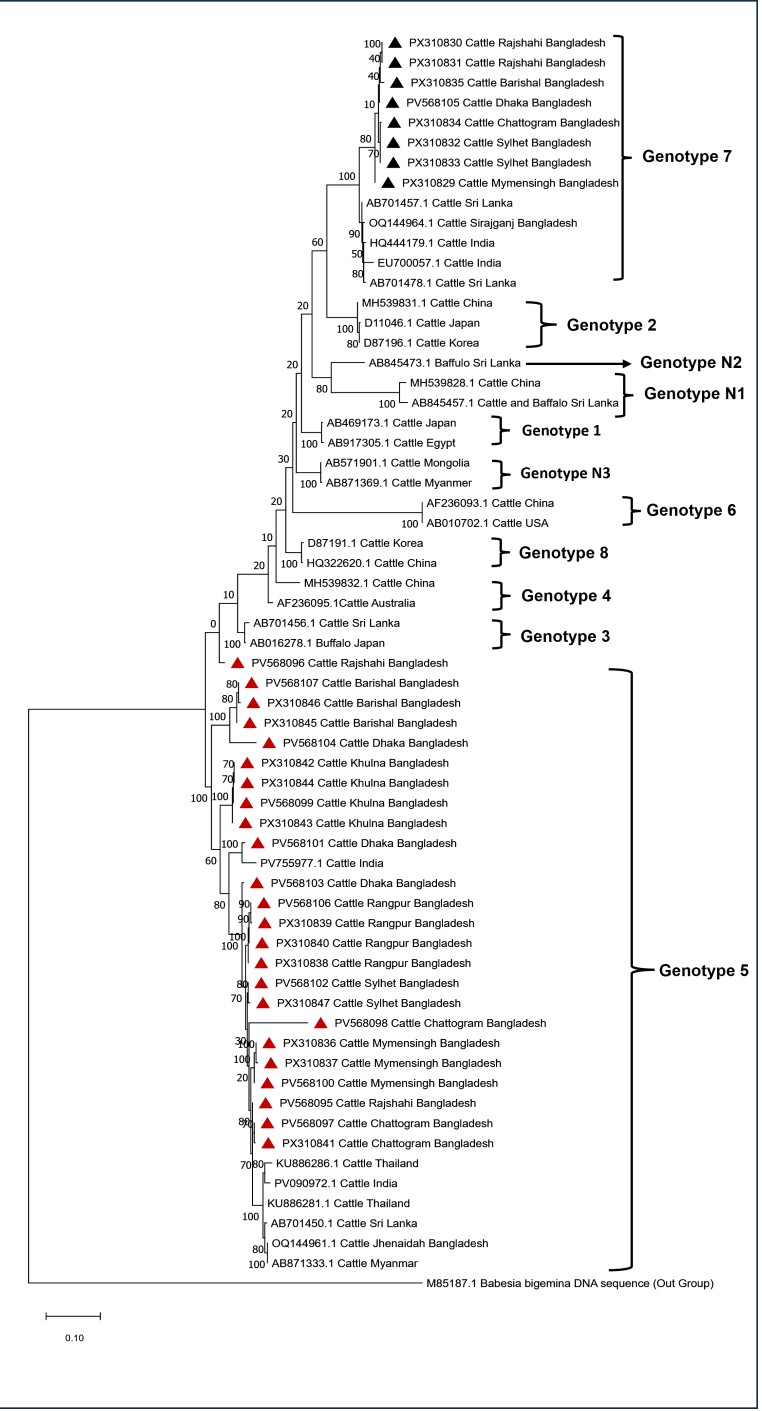

**Fig 2. Maximum-likelihood phylogenetic tree of *Theileria orientalis* MPSP gene sequences with *Babesia bigemina* (M85187.1) as the outgroup.** Sequences from this study are marked with triangles (▲ black = Genotype 7; ▲ red = Genotype 5). Labels include accession numbers (in parentheses), host, country and the numbers along with branches indicate the bootstrap values.

**Table 4. Haplotype-nucleotide diversity and neutrality tests of MPSP gene sequences of *T. orientalis*.**

| Location | n | Hn | Hd ± SD | π ± SD | K | S | Tajima's D | Fu-Li's F |
|---|---|---|---|---|---|---|---|---|
| | | | Haplotype-Nucleotide Diversity | | | | Neutrality test | |
| Mymensingh | 4 | 4 | 1 ± 0.177 | 0.08235 ± 0.041 | 62.5 | 124 | −0.7953 | −0.85353 |
| Rajshahi | 4 | 3 | 0.833 ± 0.222 | 0.112 ± 0.029 | 85.67 | 141 | 0.14557 | 0.39418 |
| Sylhet | 4 | 3 | 0.833 ± 0.222 | 0.096 ± 0.029 | 73 | 110 | 2.26785* | 2.43164** |
| Chattogram | 4 | 4 | 1 ± 0.177 | 0.120 ± 0.04 | 97.83 | 177 | −0.89133*** | −0.85158 |
| Barishal | 4 | 4 | 1 ± 0.311 | 0.082 ± 0.042 | 62.5 | 124 | −0.87275 | −0.91422 |
| Rangpur | 4 | 4 | 1 ± 0.177 | 0.003 ± 0.001 | 2.83 | 5 | 0.37186 | 0.35139 |
| Khulna | 4 | 3 | 0.833 ± 0.222 | 0.001 ± 0.004 | 1.167 | 2 | 0.59158 | 0.50356 |
| Dhaka | 4 | 4 | 1 ± 0.177 | 0.113 ± 0.02 | 86 | 154 | −0.64486 | −0.45854 |
| Total | 32 | 29 | 0.994 ± 0.009 | 0.09516 ± 0.007 | 72.22782 | 264 | −0.0537 | −0.4637 |

Note: n: Number of sequences, Hn: Number of Haplotypes, Hd: Haplotype (gene) diversity, π: Nucleotide diversity (per site) [58], K: Average number of nucleotide differences between two randomly chosen sequences from within in the population [59], S: Number of variable/segregating sites., Statistical significance: * = p < 0.05; ** = p < 0.01; *** = p < 0.001

radiating from central nodes suggests recent population expansion, with a balance between widespread connectivity and localized differentiation shaping the observed haplotype distribution (Table 7).

## Discussion

As a widespread of tick-borne disease, oriental theileriosis represents a serious threat to the sustainable livestock production across the world [63]. Comprehensive epidemiological research is crucial to elucidate the complex relationships among ticks, hosts and pathogens as well as to determine mechanisms of transmission, prevalence and pathogenesis, thereby supporting the development of new intervention strategies [64]. The presence of *T. orientalis* in Bangladesh has been established, yet details genotypic variation remain insufficient. The study highlights the prevalence and genetic diversity of *T. orientalis* based on molecular approaches.

The present study revealed an overall prevalence 63.75% of *T. orientalis* in Bangladesh. It is consistent with previous epidemiological reports in which a prevalence of 67.24% was recorded from different regions of the country [20]. In contrast, the current results are higher from the earlier reports where prevalence were recorded 7.7% in Sirajganj, 20.4% in Rangpur, 24.13% in Jhenaidah, 42.86% in Khagrachari, 50% in Bogura and 55.2% in Mymensingh districts under different divisions of Bangladesh [18–20]. The prevalence *T. orientalis* is also higher than those are reported in several other Asian countries including Myanmar (36.2%), Thailand (36.50%), Malaysia (49.76%), China (36.5%), Korea (41.3%), and Sri Lanka (53.5%) [24–26,29,30,65]. Compare to the clinical cases from the neighbouring countries, the prevalence is lower than those reported in India, where incidence rate was observed 70% in Odisha [22] and 93.3% in Himachal Pradesh [21]. The regional prevalence of *T. orientalis* is significantly (p < 0.001) higher in Chattogram, Barishal, Sylhet and Mymensingh divisions compared to other regions of the study areas. This variation might be due to high number of tick population in particular the *Haemaphysalis bispinosa* which may act a vector for transmission of *T. orientalis* [20]. The geography of Chattogram and Sylhet divisions are hilly areas that might provide favourable conditions for the propagation of tick vectors due to high humidity, rainfall and vegetation than other parts of the country [66–68]. In addition, those regions might be more exposed with wild animals and have a possibility to transfer TBDs to the cattle [26]. On the other hand, cattle in Barishal and Mymensingh divisions showed higher infection rates as animals are mostly raised in free-range conditions in the Char areas which provide enough time for grazing thus allow more tick infestation and leads to high prevalence of tick-borne pathogens (TBPs). Recently, Bangladesh has imported live cattle of various breeds such as Holstein Friesian, Brahman, Sahiwal, Jersey, Niloy,

**Table 5. Gene flow and genetic differentiation indices between two genotypes of *T. orientalis* estimated by MPSP gene sequence.**

| POPULATION 1 | POPULATION 2 | Kxy | Gst | Nst | Fst | Dxy | Da |
|---|---|---|---|---|---|---|---|
| Mymensingh | Rajshahi | 76.375 | 0.0435 | 0.02949 | 0.03328 | 0.1006 | 0.00335 |
| Mymensingh | Sylhet | 64.875 | 0.0435 | −0.0504 | −0.0443 | 0.0855 | −0.0038 |
| Mymensingh | Chottogram | 67.625 | 0 | −0.1509 | −0.1411 | 0.0891 | −0.0126 |
| Mymensingh | Barishal | 88 | 0 | 0.27498 | 0.28977 | 0.1159 | 0.0336 |
| Mymensingh | Rangpur | 39.813 | 0 | 0.15432 | 0.17949 | 0.0525 | 0.00941 |
| Mymensingh | Khulna | 71.5 | 0.0435 | 0.53917 | 0.55478 | 0.0942 | 0.05226 |
| Mymensingh | Dhaka | 70.313 | 0 | −0.0676 | −0.056 | 0.0926 | −0.0052 |
| Rajshahi | Sylhet | 66.375 | 0.0909 | −0.1987 | −0.1915 | 0.0875 | −0.0167 |
| Rajshahi | Chottogram | 87.625 | 0.0435 | −0.0004 | −0.01 | 0.1155 | −0.0012 |
| Rajshahi | Barishal | 84.688 | 0.0435 | 0.12035 | 0.12817 | 0.1116 | 0.0143 |
| Rajshahi | Rangpur | 75.75 | 0.0435 | 0.41634 | 0.41914 | 0.0998 | 0.04183 |
| Rajshahi | Khulna | 84 | 0.0909 | 0.48267 | 0.48611 | 0.1107 | 0.0538 |
| Rajshahi | Dhaka | 78.125 | 0.0435 | −0.0954 | −0.0955 | 0.1029 | −0.0098 |
| Sylhet | Chottogram | 75.625 | 0.0435 | −0.084 | −0.0898 | 0.0996 | −0.009 |
| Sylhet | Barishal | 85.5 | 0.0435 | 0.19856 | 0.2076 | 0.1127 | 0.02339 |
| Sylhet | Rangpur | 59.125 | 0.0435 | 0.3556 | 0.3587 | 0.0779 | 0.02794 |
| Sylhet | Khulna | 82.25 | 0.0909 | 0.54386 | 0.54914 | 0.1084 | 0.05951 |
| Sylhet | Dhaka | 72 | 0.0435 | −0.1061 | −0.1042 | 0.0949 | −0.0099 |
| Chottogram | Barishal | 99 | 0 | 0.21419 | 0.22054 | 0.1304 | 0.02877 |
| Chottogram | Rangpur | 54.688 | 0 | 0.10515 | 0.13448 | 0.0721 | 0.00969 |
| Chottogram | Khulna | 83.5 | 0.0435 | 0.42718 | 0.44311 | 0.11 | 0.04875 |
| Chottogram | Dhaka | 82 | 0 | −0.087 | −0.0844 | 0.108 | −0.0091 |
| Barishal | Rangpur | 79.063 | 0 | 0.57296 | 0.58682 | 0.1042 | 0.06113 |
| Barishal | Khulna | 77.25 | 0.0435 | 0.57352 | 0.58792 | 0.1018 | 0.05984 |
| Barishal | Dhaka | 76.313 | 0 | 0.01425 | 0.02703 | 0.1005 | 0.00272 |
| Rangpur | Khulna | 52.5 | 0.0435 | 0.9636 | 0.9619 | 0.0692 | 0.06653 |
| Rangpur | Dhaka | 59.563 | 0 | 0.24441 | 0.25428 | 0.0785 | 0.01995 |
| Khulna | Dhaka | 71.25 | 0.0435 | 0.38141 | 0.3883 | 0.0939 | 0.03645 |

Note: Wright's F-statistics: pairwise genetic distance [60], Nst: Gene flow and population migration among populations [60,61], Kxy: Average proportion of nucleotide differences between populations. Dxy: The average number of nucleotide substitutions per site between populations [58], Da: The number of net nucleotide substitutions per site between populations [58], Gst: Genetic differentiation index based on the frequency of haplotypes [58].

and Gyr from several countries, particularly Australia, the United States and India. As these countries have reported cases of oriental theileriosis, the likelihood of the disease emerging or spreading in Bangladesh has increased. However, lower prevalence of *T. orientalis* infection was found in cattle of Dhaka and Rangpur divisions. The predispositions might be the intensive rearing system leading to restricted grazing of cattle and frequent use of acaricides thus reduced the risk of tick infestations [24,69]. Moreover, *T. orientalis* infection may also be transmitted mechanically through hematophagous insects (lice, mosquitoes, and biting flies), vertically and iatrogenically by use of contaminated needles with untrained practitioners (unauthorized practitioners) [9]. These possible causes (climatic suitability, sub-optimal control of tick vectors, free movement of cattle) warns the endemicity of oriental theileriosis as well as the emergence of more virulent genotype in livestock population of Bangladesh [70]. Future studies should include longitudinal surveillance, seasonal sampling, and tick vector analysis to address current limitations and better understand the transmission dynamics and genetic diversity of *T. orientalis* in Bangladesh.

The NCBI blasts search and sequence alignment results revealed a high degree of genetic similarity among *T. orientalis* genotypes detected in this study and other countries of Asia. Sequences are generated in this study (PV568095 and PX310841 from Rajshahi and Chattogram divisions respectively) exhibit strangely high identity (98.02% and 97.93%) with genotypes reported from Thailand (KU886281.1) and Sri Lanka (AB701450.1) respectively, indicating a close relationship between these sequences and the isolate PV568097 from Chattogram shows substantial similarity (97.87%) with the isolates reported previously from Malawi (OP839193.1). On the other hand, sequences from Mymensingh (PV568100) and Rangpur (PV568106, PX310838) regions shared genetic identity with the genotypes reported from Thailand (KU886281.1) and Myanmar (AB871353.1) 97.10% and 96.97% respectively. The present findings accentuated the genetic diversity of *T. orientalis* within the countries of South Asia suggesting a potential regional transmission pathways and evolutionary connections. The intra-sequence similarity analysis revealed a high identity (100%) between multiple pairs indicating they likely share a common ancestor and this is further supported by the phylogenetic analysis, where sequences were clustered within the same clade. This could be signified the ongoing genetic divergence within the genotypes, potentially influenced by geographic or environmental factors that might affect the pathogen's transmission dynamics, virulence or susceptibility to control measures [71].

The genetic diversity of *T. orientalis* species has been explored through MPSP gene sequencing and documented 11 genotypes (types 1–8 and types N1–N3) from various countries of the world [43,72]. The most pathogenic genotype of *T. orientalis* belong to the Ikeda group (genotypes 2 and 7) has been previously reported from different countries, like USA, Australia, Japan, New Zealand and Bangladesh [8,20,24,43,73]. However, mixed infection with diverse *T. orientalis* genotypes have also been detected in cattle population from various Asian countries include Japan [74], Thailand [75] and Korea [15]. In the present study, the phylogenetic analysis of MPSP gene sequences identified two distinct separate genotypes, referred to as genotype 5 and genotype 7. The findings are consistent with the previous study on molecular epidemiology of *T. orientalis* in cattle of Bangladesh [20]. However, the concurrent presence of both genotypes across several divisions of Bangladeshi adds an important layer of epidemiological context, thus suggests that these genotypes are well-distributed and established within local cattle populations and may also be maintained through both vector abundance and cattle movement between regions. This finding underscores the potential for broader dissemination of *T. orientalis* within Bangladesh and reinforces the importance of ongoing molecular surveillance to monitor the genotype distribution, emerging variants, and guidance the targeted control strategies. The phenomena of clustering 24 isolates of *T. orientalis* belongs to genotype 5 from various parts of Bangladesh likely reflects its dominance due to various factors such as ecological suitability, host adaptation and potential vector specificity. This genotype may have higher transmission efficiency, adaptability to local cattle breeds or resilience against environmental stressors and control measures. Additionally, the movement of cattle across the regions for trade or farming could facilitate its widespread distribution.

The phylogenetic analysis underscores the evolutionary relationships of Bangladeshi *T. orientalis* with regional isolates, revealing that both genotype 5 and genotype 7 share close ancestry with sequences from India, Bangladesh and Sri Lanka. The well-supported sub-branches within each genotype (Genotype 5-PV568095, PV568098, PV568103, PV568096; Genotype 7- PX310829) indicate ongoing lineage diversification, suggesting independent evolutionary trajectories within Bangladesh. This structure reflects historical connectivity with neighboring regions while also highlighting local divergence, emphasizing the complex population dynamics and the potential for region-specific adaptation.

The *T. orientalis* population in Bangladesh exhibits high haplotype diversity and variable nucleotide divergence, reflecting a dynamic evolutionary history shaped by host movement, vector ecology and repeated introductions from diverse sources [76,77]. Neutrality tests reveal contrasting demographic and selective pressures across regions: negative values suggest recent population expansion or purifying selection, whereas positive values in Sylhet indicate balancing selection or population contraction. These patterns indicate a structured population, with some regions acting as reservoirs of deep genetic diversity and others showing signs of founder effects or restricted transmission, highlighting the need for region-specific surveillance and control strategies [78].

Population differentiation analyses further support this structure. While most divisions show low to moderate differentiation, suggesting gene flow or recent shared ancestry, certain population pairs, particularly Rangpur–Khulna and Barishal–Rangpur/Khulna, are highly differentiated, likely reflecting long-term isolation or local adaptation. Near-zero Nst and low Da values in other comparisons highlight connectivity and shared haplotypes, indicating that gene flow and recent expansion contribute to homogenization among some populations. Together, these results depict a mosaic of regional isolation and connectivity, shaped by ecological and host-mediated factors.

Phylogenetic analysis of the MPSP gene sequences is insufficiently informative for detecting minute genetic variations. However, haplotype network analysis has become a widely used method for uncovering genetic diversity

**Table 6. Estimation of codon-based evolutionary divergence between MPSP sequences.**

| Genotype | Seq no | | 1 | 2 | 3 | 4 | 5 | 6 | 7 | 8 | 9 | 10 | 11 | 12 | 13 | 14 |
|---|---|---|---|---|---|---|---|---|---|---|---|---|---|---|---|---|
| 5 | 1 | PV568100 | | | | | | | | | | | | | | |
| 5 | 2 | PX310836 | 0.000 | | | | | | | | | | | | | |
| 5 | 3 | PX310837 | 0.000 | 0.000 | | | | | | | | | | | | |
| 5 | 4 | PV568095 | 0.024 | 0.024 | 0.024 | | | | | | | | | | | |
| 5 | 5 | PV568096 | 0.324 | 0.323 | 0.323 | 0.362 | | | | | | | | | | |
| 5 | 6 | PV568102 | 0.012 | 0.012 | 0.012 | 0.012 | 0.343 | | | | | | | | | |
| 5 | 7 | PX310847 | 0.012 | 0.012 | 0.012 | 0.012 | 0.343 | 0.000 | | | | | | | | |
| 5 | 8 | PV568097 | 0.031 | 0.031 | 0.031 | 0.006 | 0.372 | 0.018 | 0.018 | | | | | | | |
| 5 | 9 | PV568098 | 0.131 | 0.117 | 0.113 | 0.117 | 0.422 | 0.117 | 0.117 | 0.124 | | | | | | |
| 5 | 10 | PX310841 | 0.031 | 0.031 | 0.031 | 0.006 | 0.372 | 0.018 | 0.018 | 0.000 | 0.124 | | | | | |
| 5 | 11 | PV568107 | 0.180 | 0.172 | 0.176 | 0.195 | 0.189 | 0.196 | 0.196 | 0.203 | 0.255 | 0.203 | | | | |
| 5 | 12 | PX310845 | 0.195 | 0.187 | 0.191 | 0.211 | 0.174 | 0.211 | 0.211 | 0.219 | 0.264 | 0.219 | 0.012 | | | |
| 5 | 13 | PX310846 | 0.187 | 0.179 | 0.183 | 0.203 | 0.197 | 0.203 | 0.203 | 0.211 | 0.246 | 0.211 | 0.006 | 0.018 | | |
| 5 | 14 | PV568106 | 0.018 | 0.018 | 0.018 | 0.018 | 0.339 | 0.006 | 0.006 | 0.024 | 0.132 | 0.024 | 0.188 | 0.204 | 0.195 | |
| 5 | 15 | PX310838 | 0.006 | 0.006 | 0.006 | 0.018 | 0.339 | 0.006 | 0.006 | 0.024 | 0.132 | 0.024 | 0.188 | 0.204 | 0.195 | 0.012 |
| 5 | 16 | PX310839 | 0.018 | 0.018 | 0.018 | 0.018 | 0.339 | 0.006 | 0.006 | 0.024 | 0.132 | 0.024 | 0.188 | 0.204 | 0.195 | 0.000 |
| 5 | 17 | PX310840 | 0.018 | 0.018 | 0.018 | 0.018 | 0.339 | 0.006 | 0.006 | 0.024 | 0.132 | 0.024 | 0.188 | 0.204 | 0.195 | 0.000 |
| 5 | 18 | PV568099 | 0.170 | 0.170 | 0.170 | 0.185 | 0.155 | 0.186 | 0.186 | 0.193 | 0.255 | 0.193 | 0.145 | 0.131 | 0.139 | 0.194 |
| 5 | 19 | PX310842 | 0.170 | 0.170 | 0.170 | 0.185 | 0.155 | 0.186 | 0.186 | 0.193 | 0.255 | 0.193 | 0.145 | 0.131 | 0.139 | 0.194 |
| 5 | 20 | PX310843 | 0.163 | 0.163 | 0.162 | 0.178 | 0.148 | 0.178 | 0.178 | 0.185 | 0.247 | 0.185 | 0.138 | 0.124 | 0.132 | 0.186 |
| 5 | 21 | PX310844 | 0.170 | 0.170 | 0.170 | 0.185 | 0.155 | 0.186 | 0.186 | 0.193 | 0.255 | 0.193 | 0.145 | 0.131 | 0.139 | 0.194 |
| 5 | 22 | PV568101 | 0.130 | 0.130 | 0.130 | 0.144 | 0.221 | 0.145 | 0.145 | 0.152 | 0.203 | 0.152 | 0.096 | 0.103 | 0.103 | 0.152 |
| 5 | 23 | PV568103 | 0.037 | 0.037 | 0.037 | 0.037 | 0.319 | 0.037 | 0.037 | 0.043 | 0.131 | 0.043 | 0.184 | 0.184 | 0.191 | 0.043 |
| 5 | 24 | PV568104 | 0.188 | 0.176 | 0.180 | 0.204 | 0.257 | 0.204 | 0.204 | 0.208 | 0.260 | 0.208 | 0.101 | 0.101 | 0.108 | 0.196 |
| 7 | 25 | PX310829 | 0.575 | 0.575 | 0.574 | 0.538 | 0.475 | 0.563 | 0.563 | 0.538 | 0.659 | 0.538 | 0.524 | 0.512 | 0.535 | 0.558 |
| 7 | 26 | PX310830 | 0.532 | 0.531 | 0.531 | 0.496 | 0.442 | 0.521 | 0.521 | 0.496 | 0.618 | 0.496 | 0.483 | 0.472 | 0.493 | 0.516 |
| 7 | 27 | PX310831 | 0.532 | 0.531 | 0.531 | 0.496 | 0.442 | 0.521 | 0.521 | 0.496 | 0.618 | 0.496 | 0.483 | 0.472 | 0.493 | 0.516 |
| 7 | 28 | PX310832 | 0.538 | 0.537 | 0.536 | 0.502 | 0.455 | 0.526 | 0.526 | 0.502 | 0.625 | 0.502 | 0.496 | 0.485 | 0.507 | 0.523 |
| 7 | 29 | PX310833 | 0.538 | 0.537 | 0.536 | 0.502 | 0.455 | 0.526 | 0.526 | 0.502 | 0.625 | 0.502 | 0.496 | 0.485 | 0.507 | 0.523 |
| 7 | 30 | PX310834 | 0.538 | 0.537 | 0.536 | 0.502 | 0.455 | 0.526 | 0.526 | 0.502 | 0.625 | 0.502 | 0.496 | 0.485 | 0.507 | 0.523 |
| 7 | 31 | PX310835 | 0.560 | 0.560 | 0.559 | 0.523 | 0.467 | 0.549 | 0.549 | 0.523 | 0.650 | 0.523 | 0.510 | 0.498 | 0.520 | 0.543 |
| 7 | 32 | PV568105 | 0.534 | 0.533 | 0.532 | 0.498 | 0.444 | 0.522 | 0.522 | 0.498 | 0.620 | 0.498 | 0.485 | 0.473 | 0.495 | 0.517 |

Note: The number of synonymous substitutions per synonymous site from between sequences are shown. Analyses were conducted using the Nei-Gojobori model [62]. This analysis involved 32 nucleotide sequences. All ambiguous positions were removed for each sequence pair (pairwise deletion option). There was a total of 253 positions in the final dataset. Evolutionary analyses were conducted in MEGA11 [52].

and relationships, even within a single phylogenetic lineage, through the identification of single nucleotide variations [79]. The genotype-specific patterns and haplotype network analyses reinforce the others genetic findings of the present study. The median-joining network (MJN) shows central haplotypes giving rise to multiple variants, consistent with population growth, while peripheral haplotypes indicate regional differentiation and partial isolation. The distribution of diverse haplotypes within one geographical region might permit interbreeding and subsequently elevate nuclear gene recombination during sexual reproduction in tick vectors [80,81]. Genotype 5 exhibits low divergence and strong sequence conservation, consistent with recent expansion and purifying selection, whereas Genotype 7 is more heterogeneous, reflecting longer-term evolution and localized diversification [82]. In this study,

| 15 | 16 | 17 | 18 | 19 | 20 | 21 | 22 | 23 | 24 | 25 | 26 | 27 | 28 | 29 | 30 | 31 | 32 |
|---|---|---|---|---|---|---|---|---|---|---|---|---|---|---|---|---|---|
| 0.012 | | | | | | | | | | | | | | | | | |
| 0.012 | 0.000 | | | | | | | | | | | | | | | | |
| 0.178 | 0.194 | 0.194 | | | | | | | | | | | | | | | |
| 0.178 | 0.194 | 0.194 | 0.000 | | | | | | | | | | | | | | |
| 0.171 | 0.186 | 0.186 | 0.006 | 0.006 | | | | | | | | | | | | | |
| 0.178 | 0.194 | 0.194 | 0.000 | 0.000 | 0.006 | | | | | | | | | | | | |
| 0.137 | 0.152 | 0.152 | 0.124 | 0.124 | 0.117 | 0.124 | | | | | | | | | | | |
| 0.031 | 0.043 | 0.043 | 0.155 | 0.155 | 0.148 | 0.155 | 0.123 | | | | | | | | | | |
| 0.196 | 0.196 | 0.196 | 0.244 | 0.244 | 0.235 | 0.244 | 0.157 | 0.224 | | | | | | | | | |
| 0.584 | 0.558 | 0.558 | 0.543 | 0.543 | 0.531 | 0.543 | 0.492 | 0.543 | 0.472 | | | | | | | | |
| 0.540 | 0.516 | 0.516 | 0.508 | 0.508 | 0.496 | 0.508 | 0.454 | 0.502 | 0.434 | 0.018 | | | | | | | |
| 0.540 | 0.516 | 0.516 | 0.508 | 0.508 | 0.496 | 0.508 | 0.454 | 0.502 | 0.434 | 0.018 | 0.000 | | | | | | |
| 0.546 | 0.523 | 0.521 | 0.521 | 0.521 | 0.509 | 0.521 | 0.460 | 0.507 | 0.446 | 0.025 | 0.006 | 0.006 | | | | | |
| 0.546 | 0.523 | 0.521 | 0.521 | 0.521 | 0.509 | 0.521 | 0.460 | 0.507 | 0.446 | 0.025 | 0.006 | 0.006 | 0.000 | | | | |
| 0.546 | 0.523 | 0.521 | 0.521 | 0.521 | 0.509 | 0.521 | 0.460 | 0.507 | 0.446 | 0.025 | 0.006 | 0.006 | 0.000 | 0.000 | | | |
| 0.569 | 0.543 | 0.543 | 0.535 | 0.535 | 0.523 | 0.535 | 0.478 | 0.528 | 0.459 | 0.031 | 0.012 | 0.012 | 0.018 | 0.018 | 0.018 | | |
| 0.542 | 0.517 | 0.517 | 0.509 | 0.509 | 0.497 | 0.509 | 0.455 | 0.503 | 0.435 | 0.018 | 0.000 | 0.000 | 0.006 | 0.006 | 0.006 | 0.012 | |

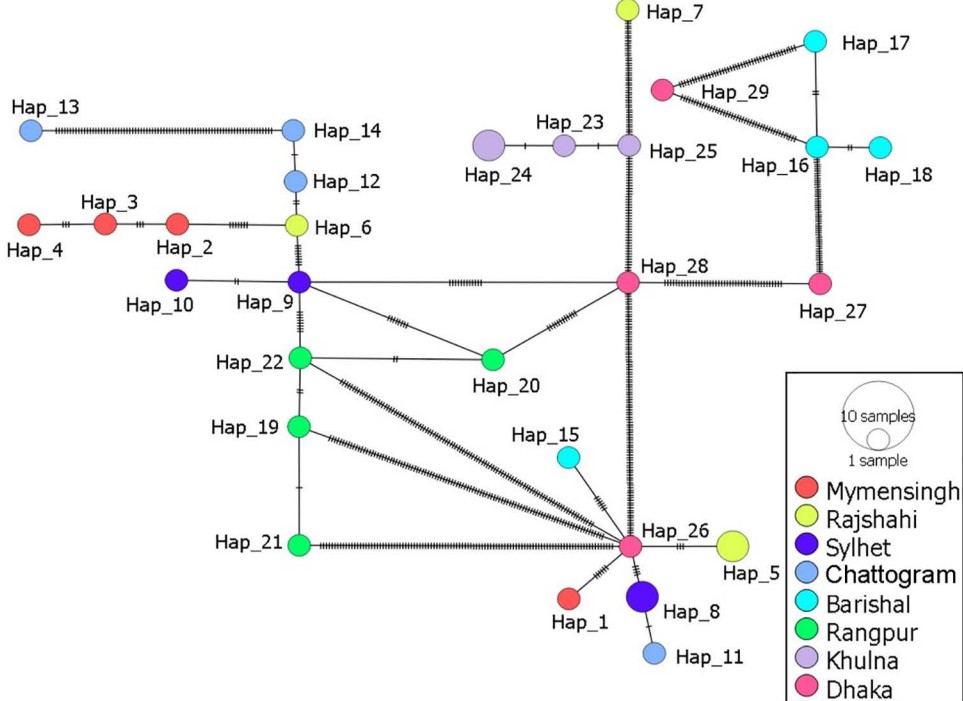

**Fig 3. Median Joining Network indicates haplotype relationships and genetic diversity of *T. orientalis* based on MPSP gene sequences.** The figure depicts a constructed Median Joining Network based on haplotype analysis of the MPSP gene sequences. Each circle represents a haplotype, with the size proportional to the number of samples sharing that haplotype. The colors indicate different haplotypes, highlighting genetic differentiation among groups. The hatch marks on the connecting lines between haplotypes represent the mutational steps separating them.

genotype analysis was conducted on a subset of 32 *T. orientalis*–positive samples, which were selected based on clear PCR amplicon quality and to ensure the representation from different geographic regions. Consequently, the genotype diversity identified in this study reflects only this sequenced subset and may not capture all *T. orientalis* variants circulating in the country. Broader sequencing efforts in future studies will be important for obtaining a more comprehensive picture of the genetic diversity of *T. orientalis* in cattle population of Bangladesh. Overall, the combined evidence highlights a population shaped by historical expansion, ongoing gene flow, and localized diversification, with both lineage and region-specific processes contributing to the complex genetic landscape of *T. orientalis* in Bangladesh.

## Conclusion

In conclusions, it was the first approach of nationwide assessment using bulk sample to evaluate prevalence, alongside the genetic diversity insights and population structure of *T. orientalis*. The present findings highlighted the widespread prevalence and genetic diversity of *T. orientalis* in cattle of Bangladesh reflecting the complex interplay of ecological and environmental factors. Genetic differentiation between *T. orientalis* genotypes suggests distinct evolutionary pathways, which could influence disease dynamics, virulence, and transmission. The findings of this study will provide awareness for all stakeholders including veterinarians and field practitioners, highlighting the increasing prevalence and genetic variations of *T. orientalis*. Furthermore, ongoing molecular surveillance and whole genome studies will be vital to understanding the evolving landscape of *T. orientalis* and improving ability to mitigate its impact through targeted interventions, including vaccines and therapeutic measures.

**Table 7. List of *Theileria orientalis* haplotypes identified based on MPSP gene sequences in cattle from eight divisions of Bangladesh.** The table includes haplotype number, number of sequences, genotypes, geographic location and their corresponding GenBank accession numbers.

| Haplotype No | No. of Seq | Genotype | Geographical location | Accession no |
|---|---|---|---|---|
| Hap_1, | 1 | 7 | Mymensingh | PX310829 |
| Hap_2, | 1 | 5 | Mymensingh | PV568100 |
| Hap_3, | 1 | 5 | Mymensingh | PX310836 |
| Hap_4, | 1 | 5 | Mymensingh | PX310837 |
| Hap_5, | 2 | 7 | Rajshahi | PX310830, PX310831 |
| Hap_6, | 1 | 5 | Rajshahi | PV568095 |
| Hap_7, | 1 | 5 | Rajshahi | PV568096 |
| Hap_8, | 2 | 7 | Sylhet | PX310832, PX310833 |
| Hap_9, | 1 | 5 | Sylhet | PV568102 |
| Hap_10, | 1 | 5 | Sylhet | PX310847 |
| Hap_11, | 1 | 7 | Chottogram | PX310834 |
| Hap_12, | 1 | 5 | Chottogram | PV568097 |
| Hap_13, | 1 | 5 | Chottogram | PV568098 |
| Hap_14, | 1 | 5 | Chottogram | PX310841 |
| Hap_15, | 1 | 7 | Barishal | PX310835 |
| Hap_16, | 1 | 5 | Barishal | PV568107 |
| Hap_17, | 1 | 5 | Barishal | PX310845 |
| Hap_18, | 1 | 5 | Barishal | PX310846 |
| Hap_19, | 1 | 5 | Rangpur | PV568106 |
| Hap_20, | 1 | 5 | Rangpur | PX310838 |
| Hap_21, | 1 | 5 | Rangpur | PX310839 |
| Hap_22, | 1 | 5 | Rangpur | PX310840 |
| Hap_23, | 1 | 5 | Khulna | PV568099 |
| Hap_24, | 2 | 5 | Khulna | PX310842, PX310843 |
| Hap_25, | 1 | 5 | Khulna | PX310844 |
| Hap_26, | 1 | 7 | Dhaka | PV568105 |
| Hap_27, | 1 | 5 | Dhaka | PV568101 |
| Hap_28, | 1 | 5 | Dhaka | PV568103 |
| Hap_29, | 1 | 5 | Dhaka | PV568104 |

## Supporting information

**S1 Fig. PCR showing specific MPSP gene amplification of 776 bp size of *Theileria orientalis*.** Lane L: 100 bp DNA ladder, Lane S1-S11: Test samples and Lane N: Negative control. All data and analysis scripts supporting this study are archived and available at https://doi.org/10.5281/zenodo.17191643
(TIF)

## Author contributions

**Conceptualization:** Md Hasanuzzaman Talukder, Babul Chandra Roy.

**Data curation:** Md Hasanuzzaman Talukder, Mostak Ahmed, Babul Chandra Roy, Md. Mahfuzur Rahman Sajib, Md. Rajiur Rahaman Rabbi, Md. Khalilur Rahman, Md. Abu Haris Miah.

**Formal analysis:** Md Hasanuzzaman Talukder, Mostak Ahmed, Babul Chandra Roy, Md. Mahfuzur Rahman Sajib, Md. Rajiur Rahaman Rabbi, Md. Makshuder Rahman Zim, Md. Abu Haris Miah, Peru Gopal Biswas.

**Funding acquisition:** Md Hasanuzzaman Talukder.

**Investigation:** Md Hasanuzzaman Talukder, Mostak Ahmed, Babul Chandra Roy, Md. Rajiur Rahaman Rabbi, Md. Makshuder Rahman Zim, Md. Khalilur Rahman, Md. Abu Haris Miah.

**Methodology:** Md Hasanuzzaman Talukder, Mostak Ahmed, Babul Chandra Roy, Md. Mahfuzur Rahman Sajib, Md. Rajiur Rahaman Rabbi, Md. Makshuder Rahman Zim, Md. Khalilur Rahman, Md. Abu Haris Miah, Peru Gopal Biswas.

**Project administration:** MD HASANUZZAMAN TALUKDER, Babul Chandra Roy.

**Resources:** Md Hasanuzzaman Talukder, Mostak Ahmed, Babul Chandra Roy, Md. Rajiur Rahaman Rabbi, Md. Makshuder Rahman Zim, Md. Khalilur Rahman, Md. Abu Haris Miah, Peru Gopal Biswas.

**Software:** Md Hasanuzzaman Talukder, Mostak Ahmed, Babul Chandra Roy, Md. Mahfuzur Rahman Sajib, Md. Rajiur Rahaman Rabbi, Md. Makshuder Rahman Zim.

**Supervision:** Md Hasanuzzaman Talukder, Babul Chandra Roy.

**Validation:** Md Hasanuzzaman Talukder, Mostak Ahmed, Babul Chandra Roy, Md. Mahfuzur Rahman Sajib, Md. Rajiur Rahaman Rabbi, Md. Makshuder Rahman Zim, Peru Gopal Biswas.

**Visualization:** Md Hasanuzzaman Talukder, Mostak Ahmed, Babul Chandra Roy, Md. Mahfuzur Rahman Sajib, Md. Rajiur Rahaman Rabbi, Md. Makshuder Rahman Zim, Md. Khalilur Rahman, Md. Abu Haris Miah, Peru Gopal Biswas.

**Writing – original draft:** Md Hasanuzzaman Talukder, Mostak Ahmed, Babul Chandra Roy, Md. Rajiur Rahaman Rabbi.

**Writing – review & editing:** Md Hasanuzzaman Talukder, Mostak Ahmed, Babul Chandra Roy, Md. Mahfuzur Rahman Sajib, Md. Rajiur Rahaman Rabbi, Md. Makshuder Rahman Zim, Md. Khalilur Rahman, Md. Abu Haris Miah.

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
