## [Decision Letter · Decision Letter 0]

26 Nov 2025

Dear Dr. TALUKDER,

We look forward to receiving your revised manuscript.

Kind regards,

Shahin Tajeri, D.V.M. Ph.D.

Academic Editor

PLOS ONE

Journal Requirements:

“The authors acknowledge the financial support (Principal investigator, M. H. T.; research and innovation subproject; project code: (RP-C-03-24) from the Livestock and Dairy Development Project (LDDP), jointly funded by the World Bank and Department of Livestock Services (DLS), Ministry of Fisheries and Livestock, Govt. of Bangladesh.”

3. Please note that your Data Availability Statement is currently missing the repository name. If your manuscript is accepted for publication, you will be asked to provide these details on a very short timeline. We therefore suggest that you provide this information now, though we will not hold up the peer review process if you are unable.

5. We note that Figure 1 in your submission contain map images which may be copyrighted. All PLOS content is published under the Creative Commons Attribution License (CC BY 4.0), which means that the manuscript, images, and Supporting Information files will be freely available online, and any third party is permitted to access, download, copy, distribute, and use these materials in any way, even commercially, with proper attribution. For these reasons, we cannot publish previously copyrighted maps or satellite images created using proprietary data, such as Google software (Google Maps, Street View, and Earth). For more information, see our copyright guidelines: http://journals.plos.org/plosone/s/licenses-and-copyright.

Additional Editor Comments:

When revising your manuscript please consider below comments plus those of the #reviewer 1:

Lines 35-36: Please change the sentence to: *“In contrast, **T. orientalis** does induce lymphoproliferation.”*Lines 36-37: Please provide a reference for the tick vectors. I do not believe all tick species listed act as vectors.Please remove the abbreviation “Md.” from the author names.Please delete the **Acknowledgments**  section, as it is not relevant.Please ensure that the bibliography follows **PLOS ONE**  formatting guidelines.Since the study focuses on **T. orientalis** , which causes a disease distinct from other *Theileria*  species, please be specific throughout the manuscript. Avoid using the general term “theileriosis” for *T. orientalis*  infection (e.g., line 236).

Reviewers' comments:

Reviewer's Responses to Questions

**Comments to the Author**

1. Is the manuscript technically sound, and do the data support the conclusions?

Reviewer #1: Yes

2. Has the statistical analysis been performed appropriately and rigorously?

Reviewer #1: I Don't Know

3. Have the authors made all data underlying the findings in their manuscript fully available?

Reviewer #1: Yes

4. Is the manuscript presented in an intelligible fashion and written in standard English?

Reviewer #1: Yes

Reviewer #1: Thank you for the opportunity to review your manuscript. This is an original and well-structured study reporting the molecular detection and genetic diversity of Theileria orientalis in cattle across Bangladesh. The large sample size, national coverage, and sequence confirmation make this work a valuable contribution to regional epidemiology and tick-borne disease surveillance. Overall, the study is technically sound, conclusions are supported by the data, and the English is clear. Only minor revisions are needed to enhance clarity and reporting consistency.

1. Sampling description

The random sampling strategy is appropriate and ensures diversity of age, breed, and sex. To enhance transparency, please briefly describe how randomization was implemented within farms, herds, or markets (e.g., random selection on site).

2. PCR methodology

It is clear that positives were confirmed by sequencing, which supports assay specificity. Please indicate explicitly whether no-template or extraction controls were included.

Although homologous MPSP-like genes exist in T. parva and T. annulata, the primer pair used here targets a region unique to the T. orientalis complex (as validated in Kamau et al., 2011). Mentioning this would clarify assay specificity given the absence of local DNA from other Theileria species in Bangladesh.

3. Statistical reporting

The use of Pearson’s chi-square and Fisher’s exact tests (SPSS v27) is appropriate for the descriptive objective, and this information is already clearly stated in the Methods. Reporting 95 % confidence intervals for prevalence estimates would enhance interpretability, but no further analysis is required.

4. Sequencing subset

A total of 510 bovines' blood were found positive with T. orientalis. Could you please state briefly how the 32 sequenced isolates were selected (randomly or by gel quality/representation). Clarify that genotype diversity reflects this subset and may not capture all variants nationwide.

5. Genotype classification

The study identifies T. orientalis MPSP genotypes 5 and 7, both of which have been reported previously in Asia. While these genotypes are not novel, documenting their concurrent circulation across multiple Bangladeshi divisions adds valuable epidemiological context. The authors could emphasize this contribution more clearly in the Discussion.

6. Figures and presentation

Figure 1 (Map): Please cite the map source/shapefile and ensure boundary lines are clearly visible in print.

Figure 3 (Haplotype network): Please just verify label legibility (≥ 8 pt).

7. Discussion

The Discussion is well written and adequately contextualizes the results within regional and global T. orientalis epidemiology. The comparative analysis with previous studies is informative. However, the section could be strengthened by briefly explaining the ecological or management factors influencing regional differences, and by adding a concise concluding paragraph linking the findings to surveillance or control strategies. Overall, this is a solid and publishable discussion after minor revisions.

The manuscript is written in standard English. Minor typographical and taxonomic corrections (e.g., Amblyomma testudinarium, italicised species names, consistent “Chattogram”) can be made at proof stage.

This manuscript provides useful baseline data on T. orientalis distribution and diversity in Bangladesh. With the small clarifications above, it will fully meet PLOS ONE’s publication criteria

**Do you want your identity to be public for this peer review?** For information about this choice, including consent withdrawal, please see our Privacy Policy

Reviewer #1: No

---

## [Author Response · Author response to Decision Letter 1]

21 Dec 2025

Manuscript Title: Insights into the genetic diversity and population structure of prevalent Theileria orientalis in Bangladesh

Manuscript Number: PONE-D-25-52299

Journal: PLOS ONE

Dear Editor

We sincerely appreciate the time and effort you have taken to review our manuscript. Your insightful and constructive comments have greatly helped us to improve the quality and clarity of the paper. We have carefully revised the manuscript in accordance with your suggestions.

Below, we provide a detailed, point-by-point response to each comment.

• Reviewer comments are presented in bold.

• Our responses follow beneath each comment in regular text.

• The revised manuscript clearly indicates all changes, highlighted in light blue, and corresponding responses are provided in this rebuttal letter.

We hope that our revisions satisfactorily address all concerns and that the manuscript is now suitable for consideration in PLOS ONE.

With Kindest regards

(Professor Dr. MD H. Talukder)

on behalf of all authors

Authors’ responses to the comments related to the journal requirements

Response:

Thank you for the reviewer’s suggestion. We have carefully revised the manuscript to ensure that it fully meets the PLOS ONE style requirements. All formatting has been updated according to the PLOS ONE templates for the main body and for the title/authors/affiliation’s sections. Additionally, all files have been renamed following the journal’s file-naming guidelines.

“The authors acknowledge the financial support (Principal investigator, M. H. T.; research and innovation subproject; project code: (RP-C-03-24) from the Livestock and Dairy Development Project (LDDP), jointly funded by the World Bank and Department of Livestock Services (DLS), Ministry of Fisheries and Livestock, Govt. of Bangladesh.”

Response:

We confirm that the funders’ contribution was limited solely to financial support. They were not engaged in designing the study, data acquisition, data analysis, manuscript drafting, or publication decisions. Accordingly, the required statement has been added to the cover letter.

3. Please note that your Data Availability Statement is currently missing the repository name. If your manuscript is accepted for publication, you will be asked to provide these details on a very short timeline. We therefore suggest that you provide this information now, though we will not hold up the peer review process if you are unable.

Response:

The required data repository details have been provided in the cover letter. Please note that the link is now included as recommended. All data and analysis scripts supporting this study are archived and available at https://doi.org/10.5281/zenodo.17191643

Response:

We appreciate the reviewer’s observation. The full ethics statement has been added to the methodology section. Specifically, we have provided the complete name of the Institutional Ethics Committee that approved the study and clarified the consent procedure.

5. We note that Figure 1 in your submission contain map images which may be copyrighted. All PLOS content is published under the Creative Commons Attribution License (CC BY 4.0), which means that the manuscript, images, and Supporting Information files will be freely available online, and any third party is permitted to access, download, copy, distribute, and use these materials in any way, even commercially, with proper attribution. For these reasons, we cannot publish previously copyrighted maps or satellite images created using proprietary data, such as Google software (Google Maps, Street View, and Earth). For more information, see our copyright guidelines: http://journals.plos.org/plosone/s/licenses-and-copyright.

Response:

We appreciate the editor’s guidance regarding copyright. The previously submitted map has now been replaced with a newly generated map of the study area, created entirely by the authors. No copyrighted material was used in this new figure and therefore no permission is required. The figure caption has been updated accordingly.

6. PLOS ONE now requires that authors provide the original uncropped and unadjusted images underlying all blot or gel results reported in a submission’s figures or supporting information files. This policy and the journal’s other requirements for blot/gel reporting and figure preparation are described in detail at https://journals.plos.org/plosone/s/figures#loc-blot-and-gel-reporting-requirements and https://journals.plos.org/plosone/s/figures#loc-preparing-figures-from-image-files. When you submit your revised manuscript, please ensure that your figures adhere fully to these guidelines and provide the original underlying images for all blot or gel data reported in your submission. See the following link for instructions on providing the original image

data: https://journals.plos.org/plosone/s/figures#loc-original-images-for-blots-and-gels.

Response:

We thank the editors for this requirement. The previously included gel image has been replaced because it was cropped. We have now provided a new version of the gel image that is original, uncropped, and unadjusted, in full compliance with PLOS ONE blot/gel reporting guidelines. The corresponding original underlying image has been included in the Supporting Information.

If additional gel or blot images or further raw data are required, we would be happy to provide them.

Response:

We thank the reviewer for this important reminder. We have carefully reviewed the reference list to ensure that all citations are complete, accurate, and up to date. Any references that have been retracted have been either removed or replaced with relevant current literature. In cases where it was necessary to cite a retracted article, we have clearly indicated its retracted status in the reference list and included a citation to the retraction notice in the manuscript text. All changes to the reference list have been highlighted in the tracked changes and are noted in this rebuttal letter.

Additional Editorial Comments:

#reviewer 1:

• Lines 35-36: Please change the sentence to: “In contrast, T. orientalis does induce lymphoproliferation.”

Response:

We have revised the sentence as suggested. The revised text now reads: In contrast, T. orientalis does induce lymphoproliferation. The manuscript has been revised to include this information in the introduction section (Line 36-37).

• Lines 36-37: Please provide a reference for the tick vectors. I do not believe all tick species listed act as vectors.

Response

We thank the reviewer for pointing out this. We have carefully reviewed the literature and updated the tick vector information to include only species confirmed as vectors of Theileria orientalis. The manuscript has been revised to include this information in the introduction section (Line 37). Appropriate references have also been added.

• Please remove the abbreviation “Md.” from the author names.

Response

We thank the reviewer for the suggestion. “Md” is an integral part of the authors’ official names. Following the journal’s suggestion, we have removed the dot after “Md.”

Please delete the Acknowledgments section, as it is not relevant.

Response

Thank you for the comment. The Acknowledgments section has been removed from the revised manuscript as requested.

• Please ensure that the bibliography follows PLOS ONE formatting guidelines.

Response

We appreciate the reviewer’s observation. The entire bibliography has been checked and reformatted according to the PLOS ONE reference style, including authorship format, article titles, journal names, volume/issue numbers, page ranges, and DOI formatting.

• Since the study focuses on T. orientalis, which causes a disease distinct from other Theileria species, please be specific throughout the manuscript. Avoid using the general term “theileriosis” for T. orientalis infection (e.g., line 236).

Response

Thank you for this important clarification. We agree that Theileria orientalis infection is clinically and epidemiologically distinct from bovine theileriosis caused by T. annulata or T. parva. In the revised manuscript, we have replaced all general uses of the term “theileriosis” with the specific phrase “T. orientalis (line 16, 18, 22, 25, 36, 37, 39, 42, 47, 61, 63, 67, 69, 75, 77, 78, 111, 115, 121, 130, 135, 141, 154, 155-158, 161,174,185, 188,200, 206, 216, 223, 243, 257-259, 263, 267, 270, 279, 281, 287, 289, 296, 303, 305, 307, 310, 314, 316, 321, 328, 353, 355, 358, 362-363, 366-367) infection” or oriental theileriosis (line 45,46,60,75,81,253,278,284) (including at line 236) to ensure accuracy and avoid confusion. The manuscript has been updated accordingly (Lines 253-254, 281, 284).

Reviewer’s Comments to the Author (5)

Reviewer #1: Thank you for the opportunity to review your manuscript. This is an original and well-structured study reporting the molecular detection and genetic diversity of Theileria orientalis in cattle across Bangladesh. The large sample size, national coverage, and sequence confirmation make this work a valuable contribution to regional epidemiology and tick-borne disease surveillance. Overall, the study is technically sound, conclusions are supported by the data, and the English is clear. Only minor revisions are needed to enhance clarity and reporting consistency.

1. Sampling description

The random sampling strategy is appropriate and ensures diversity of age, breed, and sex. To enhance transparency, please briefly describe how randomization was implemented within farms, herds, or markets (e.g., random selection on site).

Response:

Thank you for the constructive comment. We agree that providing additional details improves transparency. We have now clarified the implementation of randomization in the Methodology section. Specifically, we described that within each farm/herd, animals were selected using a simple random approach. The manuscript has been revised to include this information in the Methodology section (Lines 98-99).

2. PCR methodology

It is clear that positives were confirmed by sequencing, which supports assay specificity. Please indicate explicitly whether no-template or extraction controls were included.

Although homologous MPSP-like genes exist in T. parva and T. annulata, the primer pair used here targets a region unique to the T. orientalis complex (as validated in Kamau et al., 2011). Mentioning this would clarify assay specificity given the absence of local DNA from other Theileria species in Bangladesh.

Response:

Thank you for the insightful comments. Specifically, we confirm that no-template (negative) control was included in all PCR runs. Additionally, we have added a statement regarding primer specificity. The MPSP primers target a region unique to the T. orientalis complex. Although T. annulata and T. parva possess homologous genes, this primer pair has been validated to amplify only T. orientalis, supporting the specificity of our assay in the absence of local DNA from other Theileria species. We have added this detail to the methodology section of the revised manuscript (Lines 119-123).

3. Statistical reporting

The use of Pearson’s chi-square and Fisher’s exact tests (SPSS v27) is appropriate for the descriptive objective, and this information is already clearly stated in the Methods. Reporting 95% confidence intervals for prevalence estimates would enhance interpretability, but no further analysis is required.

Response:

Thank you for this constructive suggestion. In accordance with the reviewer’s rec

---

## [Decision Letter · Decision Letter 1]

5 Jan 2026

Insights into the genetic diversity and population structure of prevalent Theileria orientalis in Bangladesh

PONE-D-25-52299R1

Dear Dr. TALUKDER,

We’re pleased to inform you that your manuscript has been judged scientifically suitable for publication and will be formally accepted for publication once it meets all outstanding technical requirements.

Kind regards,

Shahin Tajeri, D.V.M. Ph.D.

Academic Editor

PLOS One

Additional Editor Comments (optional):

Dear Dr. Talukder,

Please consider correcting line 36 to 'does not induce' when finally checking the accepted proof. Please excuse me as this was a typing error of mine when writing my review.

Regards,

Shahin Tajeri

Reviewers' comments:

Reviewer's Responses to Questions

**Comments to the Author**

Reviewer #1: All comments have been addressed

2. Is the manuscript technically sound, and do the data support the conclusions?

Reviewer #1: Yes

3. Has the statistical analysis been performed appropriately and rigorously?

Reviewer #1: Yes

4. Have the authors made all data underlying the findings in their manuscript fully available?

Reviewer #1: Yes

5. Is the manuscript presented in an intelligible fashion and written in standard English?

Reviewer #1: Yes

Reviewer #1: Thank you for the opportunity to review your manuscript. This is an original and well-structured study reporting the molecular detection and genetic diversity of Theileria orientalis in cattle across Bangladesh. The large sample size, national coverage, and sequence confirmation make this work a valuable contribution to regional epidemiology and tick-borne disease surveillance. Overall, the study is technically sound, conclusions are supported by the data, and the English is clear.

**Do you want your identity to be public for this peer review?** For information about this choice, including consent withdrawal, please see our Privacy Policy

Reviewer #1: No

---

## [Editor Report · Acceptance letter]

PONE-D-25-52299R1

PLOS One

Dear Dr. TALUKDER,

I'm pleased to inform you that your manuscript has been deemed suitable for publication in PLOS One. Congratulations! Your manuscript is now being handed over to our production team.

Kind regards,

on behalf of

Dr. Shahin Tajeri

Academic Editor

PLOS One